# Reinforcement-Learning-Based Software-Defined Edge Task Allocation Algorithm

**Tianhao Zhang \*** , **Xiaojuan Zhu and Cai Wu**

School of Computer Science and Engineering, Anhui University of Science and Technology,
Huainan 232000, China
*   Correspondence: iamzth@126.com

**Abstract:** With the rapid growth in the number of IoT devices at the edge of the network, fast, flexible and secure edge computing has emerged, but the disadvantage of the insufficient computing power of edge servers is evident when dealing with massive computing tasks. To address this situation, firstly, a software-defined edge-computing architecture (SDEC) is proposed, merging the control layer of the software-defined architecture with the edge layer of edge computing, where multiple controllers share global information about the network state through an east–west message exchange, providing global state for the collaboration of edge servers. Secondly, a reinforcement-learning-based software-defined edge task allocation algorithm (RL-SDETA) is proposed in the software-defined IoT, which enables controllers to allocate computational tasks to the most appropriate edge servers for execution based on the global network information they have obtained. Simulation results show that the RL-SDETA algorithm can effectively reduce the finding cost of the optimal edge server and reduce the task completion time and its energy consumption compared to various task allocation methods such as random and uniform.

**Keywords:** edge computing; software-defined; task allocation; reinforcement learning

## 1. Introduction

The quantity of data generated by IoT devices has increased significantly in recent years. As these IoT devices usually have limited computational power, there is a need to offload computational tasks from the resource-constrained device side to edge servers with more computational power to meet the demand for low latency and bandwidth savings [1]. However, compared to cloud servers, edge servers also have limited computing resources. As more and more data and information need to be analysed, processed and stored on edge servers, tasks can be distributed across multiple edge servers to meet the rapid processing of massive tasks. The advantage of the software-defined network (SDN) architecture is the flexibility to define and extend the functionality of the entire system [2]. Based on the characteristics of a software-defined architecture, the control layer can schedule and control the network as a whole without touching the low-level configuration [3].

Typically, applications for edge-computing runtime can be divided into multiple steps, and each step can specifically be divided into multiple tasks [4]. For the problem of edge servers' computational capacity, an operational mechanism is needed to support latency-sensitive edge-computing tasks, enabling edge servers to collaborate with each other to execute the tasks. The divided subtasks are assigned to different edge servers for execution, and the results are passed back after execution.

In this paper, computing resources are allocated on-demand through a software-defined architecture, with the aim of enabling multiple edge servers to work together on tasks in a highly automated and intelligent manner. Software-defined edge computing as an open IoT system architecture decouples upper-layer IoT applications from the underlying physical resources at the edge and builds dynamically reconfigurable intelligent

edge services [5]. The advantage of this new architecture is the ability to achieve logical and centralised control of distributed network nodes as well as IoT devices. SDN has been extensively studied in terms of controller deployment in software definition [6,7], controller scalability [8] and SDN applications [9]. However, to our knowledge, collaborative processing of edge-computing tasks in the IoT environment has been rarely addressed. To this end, this paper proposes a software-defined edge-based architecture, which allows the controller to collect information from edge servers and sense the network state from a global perspective by separating the control plane from the data plane. Based on the decisions made by the SDN controller after sensing, it determines whether the current task is to be executed on the current edge server or whether some or all of the tasks are to be distributed to other edge servers for coprocessing.

In order to minimise the task completion time and the energy consumption of the sensor network, this paper proposes a reinforcement-learning-based software-defined edge task allocation algorithm. The strategy aims to obtain an efficient solution for the computational resource allocation and task layout. In this regard, we evaluate the performance of the task allocation scheme proposed in this paper under the SDEC architecture and compare it with the random and uniform allocation schemes. The experimental results show that the scheme can effectively improve the efficiency of the task allocation and reduce the energy consumption in the late iteration. The main contributions of this paper include:

1. The synchronisation of the state information between edge servers is addressed through the east–west architecture of SDNs.
2. A software-defined edge-computing architecture is proposed. By fusing the control layer with the edge layer, global information about edge servers, network states and tasks can be obtained, thus enabling multiple edge servers to perform tasks together in an edge environment.
3. A reinforcement-learning-based edge task allocation algorithm in software-defined IoT is proposed, which can effectively reduce the cost of finding the optimal edge server, lower the task completion time and reduce energy consumption. We conduct extensive experiments to evaluate the performance of the scheme. The experimental results show that the algorithm can reduce task completion time as well as energy consumption compared to random and uniform task computation offloading.

The paper is organised as follows: Section 2 briefly discusses related work. Section 3 presents the software-defined edge-computing architecture. Section 4 presents the reinforcement-learning-based software-defined edge task allocation algorithm. We conduct extensive experiments to evaluate the algorithm and present the evaluation results in Section 5. Section 6 concludes the paper.

## 2. Related Work

### 2.1. Distributed Controller Architecture

When SDN architectures are deployed in today's real networks, some of the larger networks are usually divided into several smaller subnets, each with an SDN controller that can only store the local network view [8]. To address this, Lin et al. proposed a high-performance network view exchange mechanism for multidomain networks [10], laying the groundwork for an east–west architecture for SDNs.

The east–west architectures of SDNs can currently be divided into two main categories, namely hierarchical and horizontal architectures [11–14]. Hierarchical architectures are characterised by the fact that no east–west interfaces are erected between regional controllers, and the data interaction between them is done through an upper layer of controllers. The core of this approach is to convert the east–west interface into a north–south interface, mainly represented by the open exchange protocol (OXP), an east–west SDN architecture for SDN mobile self-assembly proposed by Yang et al. [11]. The difference between the horizontal architecture and the hierarchical architecture is that the data interaction between controllers no longer needs to be done by an upper-layer controller, but there is an east–west interface between each of them to complete the necessary data interaction. A communica-

tion interface for distributed control plane (CIDC) for a distributed control plane proposed by Benamrane et al. [12] is typical of this architecture, but this communication interface is limited to small-scale network environment; the effective west–east control association network (WECAN) [13] proposed by Yu et al. can effectively control network entities to communicate in a large-scale network environment. In addition, for the application of distributed controller framework in the IoT environment, the UbiFlow system [14] proposed by Wu et al. is oriented towards traffic control and mobility management in the IoT environment, and the system achieves distributed control of IoT traffic.

With a distributed controller architecture, the controller is able to obtain global view information in multiple SDN domains. In this paper, we use the global view information in the controller to perform reinforcement learning and find the optimal edge task allocation scheme.

### 2.2. Allocation of Edge Tasks

Task allocation and offloading is one of the main research elements in edge computing, which makes up for the shortage in computing power and storage resources of end devices and improves the ability of task processing in edge computing [15,16]. Due to the limited computing capacity of edge servers, when the edge server enters a high load state in the face of a large number of offload requests, it needs to allocate tasks to other edge servers or to federate cloud centres for processing. Based on edge cloud architectures, a number of architectures already exist to manage and coordinate computing resources at the edge and in the cloud, and the potential benefits in dynamic scenarios have been evaluated [17,18]. There are also current research directions based on three-tier offload architectures; for example, Wu et al. proposed a three-tier offload computing architecture for federating cloud centres to offload some of the edge tasks with low latency requirements to cloud centres for execution through energy latency awareness [19].

In contrast, assigning tasks to other edge servers means that the tasks are not offloaded to the cloud, and multiple edge servers at the edge are used to perform the tasks together. When one edge server is unable to handle the impact of a large number of IoT devices in a timely manner, multiple edge servers are required to work together to complete the tasks, thereby increasing the task-processing capacity at the edge [20–22]. There are two main directions from the existing task allocation schemes, which are security and energy efficiency considerations. From the perspective of security, Angelo et al. provided a trusted collaboration between edge servers through a blockchain framework, mainly from the perspective of security, to consider distributing tasks to other edge servers for processing [23]; from the perspective of energy efficiency, Ramtin et al. proposed an EEDOS collaboration-based allocation scheme for edge-computing tasks from the perspective of energy efficiency and latency awareness that greatly improved energy efficiency [24].

In addition, the global optimisation brought by SDNs also provides a lot of room for improving the effect of edge task allocation. Bassem et al. proposed a method for dynamic task scheduling based on SDNs [25], which solved the problem of resource allocation and energy awareness in edge computing by means of reinforcement learning. Zhang et al. built an SDN-based in-vehicle MEC architecture [26], which solved the task-offloading and resource allocation problems in in-vehicle networks by sensing the global state of the network and improved the operational efficiency of in-vehicle networks. From an application perspective, Jieun et al. proposed an intelligent task-offloading model for edge computing for a series of forest fire scenarios, which attempted to solve the problem of prediction and rapid response to forest fires by preventing individual edge nodes from being overloaded through a task collaboration between edge nodes and responding quickly to potential hazards in emergency situations [27].

## 3. Software-Defined Edge-Computing Architecture

The SDEC architecture proposed in this paper enables a global information transfer in the edge environment through the east–west interface of SDNs, which mainly takes advan-

tage of the SDN's real-time mastering of the topological state of the whole network and makes the collaborative processing of tasks between different edge servers more efficient, as shown in Figure 1. We extend the idea of software definition to the IoT environment using edge computing. The computing resources and service capabilities of these edge servers are virtualised and abstracted through cyberphysical mapping techniques, and ultimately, the SDEC controller enables the allocation of edge tasks.

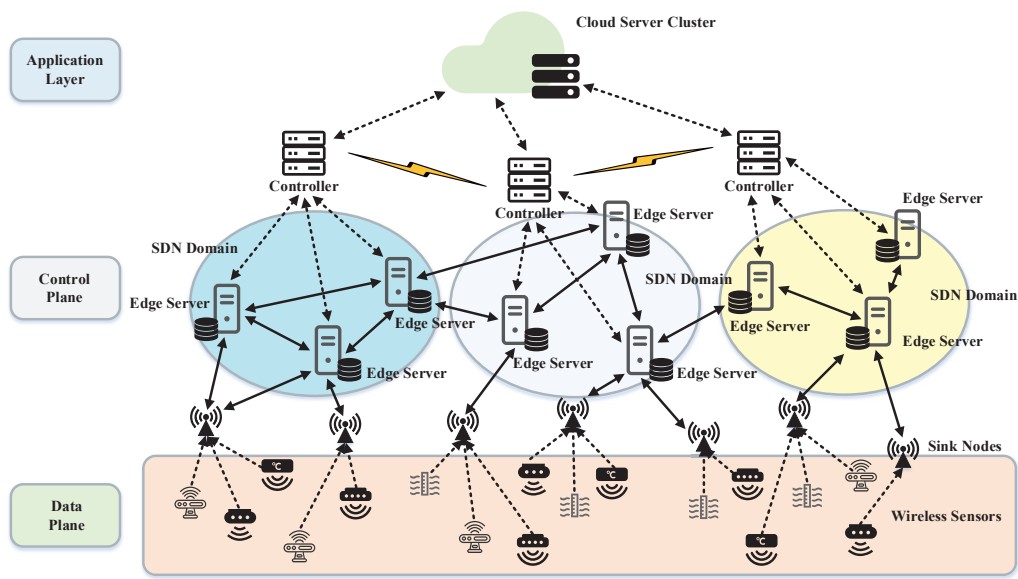

**Figure 1.** SDEC architecture. The SDEC architecture has three layers, converging the application, control and data layers of the software-defined architecture with the cloud, edge and end of the edge computing architecture.

The SDEC architecture makes full use of the east–west interface of SDNs, allowing the controllers to be connected into a network for fast information exchange. SDEC controllers are deployed at the edge layer of the edge computing architecture, allowing a global view to be shared to get the distribution of computing resources at the edge of the network.

*3.1. Network Global View*

The network view consists mainly of static and dynamic information [10]. In a software-defined architecture, the static aspects of the edge network information can be divided into three parts:

First of all, the topology information between the controllers, including the status information of the controllers, Openflow switches and wireless sensor devices, as well as the status, bandwidth and port throughput of the links. Secondly, the reachability of the edge servers: each controller is connected to an edge server in its area, and this section includes information on the status of the edge servers and their reachability. Finally, there is the quality of service (QoS) of the network, which includes the transmission delay of the network, the reliability and packet loss rate of the links, delay variation and cost.

The dynamic aspect of the edge network information consists mainly of the real-time state of the entire network architecture, such as real-time bandwidth utilisation in the topology information and all flow paths in the network. In order to avoid congestion throughout the network, the advantage of a centralised control of the SDN architecture lies in the fact that flow tables can be planned dynamically according to the specifics of the entire edge network and therefore also need to be treated as dynamic information about the network.

A global view of the entire edge network is formalised, and a unified tag is used to deliver the network global view message. The global view message contains the controller

information, link information, port information, edge server information, etc., and is stored using key–value pairs in the format shown in Table 1.

**Table 1.** Key–value table of global view storage.

| Key | Column |
| --- | --- |
| Controller_ID | IP address, Port number, System version, Edge Server_ID, Supplier name, Device type, Device function |
| Link_ID | Source Controller_ID, Destination Controller_ID, Source Port_ID, Destination Port_ID, Is_Link_Active, Bandwidth |
| Port_ID | Controller_ID, Port_MAC, Is_Active, Throughput |
| Edge Server_ID | Controller_ID, CPU model, CPU frequency, Memory type, Memory capacity, Remaining computing resources |
| Controller_Capability | Protocol name, version |
| Reachability | Edge Server IP prefixes, Length |
| Link_Utilities | Link_ID, Link utilities |

Edge network global view messages contain the controller information, link information, port information, edge server information and more. A uniform format is used and encapsulated into an XML file format for delivery, making the network global view message format flexible and easily extensible. The global network view is constructed through the interaction of global view messages between controllers.

### 3.2. Global View Information Exchange Mechanism

After the controllers have discovered each other, each controller is informed of the addresses where the other controllers are located. All controllers can then create a virtual full mesh topology to exchange and share global view information.

When exchanging global view information, messages are sent between controllers to keep the links linked and the information passed. We divided the messages between the controllers into five main categories, as shown in Table 2, and each message had a different function to support the global view information exchange.

**Table 2.** Message type table for information exchange.

| Message Type | Function |
| --- | --- |
| HELLO | The first message sent after the TCP connection is established |
| KEEPALIVE | Send at regular intervals to confirm the connection |
| VIEW-REQUEST | Request network global view |
| UPDATE | Global view update information sent to other controllers |
| ERROR | Report problems with itself or adjacent controllers to all other controllers |

The specific exchange is shown in Figure 2. The fast exchange of information between controllers requires a TCP connection to be established first. When a controller A initiates a connection to another controller B, it sends a SYN packet and waits for its acknowledgement. Controller B receives the SYN packet from controller A, acknowledges it and then ends the LISTEN phase and returns a TCP message. Controller A receives the SYN + ACK packet sent, makes it clear that the data transfer from controller A to controller B is normal, returns the acknowledgement packet ACK and controller A and controller B enter the connection state. Once a TCP connection has been established between two controllers, an OPEN message needs to be sent to confirm that the other is running; when an OPEN message is received from the other, a reply needs to be sent to confirm the OPEN message. To ensure the exchange of controller views, a KEEPALIVE message is sent continuously to confirm that the link is open. The controller will continuously send KEEPALIVE messages to neighbouring controllers via a timer to confirm the survival of the neighbouring controller. When the controller does not receive a reply to the KEEPALIVE message several times, it will assume that the neighbouring controller is not alive and pass the ERROR message to the other controllers to update their stored global view.

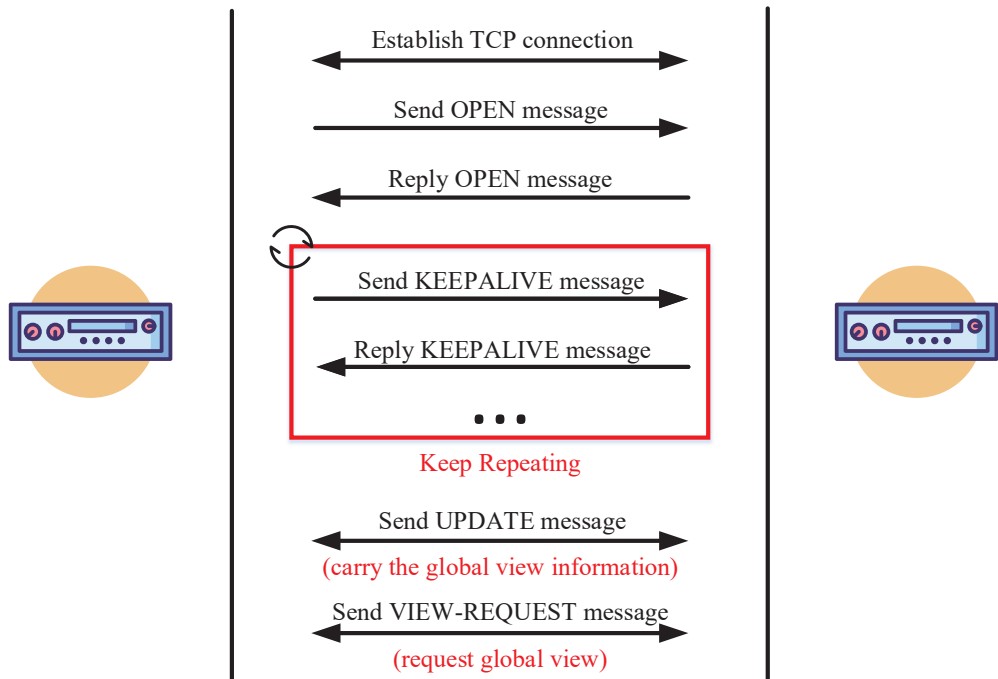

**Figure 2.** Global view exchange mechanism. SDN controllers exchange messages between them to achieve the effects of east–west connectivity, view updates, view requests, etc., and constantly communicate with each other to keep the global view up to date.

During the transfer of the network global view, updates to the global view are sent between controllers via UPDATE messages to other controllers. When a controller needs to obtain the global view of a neighbouring controller, it can request its stored network global view from the neighbouring controller by sending a VIEW-REQUEST message to the neighbouring controller.

The SDN controller can be seen as an agent while outside the agent is regarded as the environment, and environment changes can be obtained from the SDN global view. Under the SDEC architecture, controllers can be informed of the distribution of resources across the IoT edge environment in a timely manner and find the optimal edge-computing task-allocation strategy.

## 4. Optimal Edge Task Allocation Algorithm

For tasks that needed to be assigned, we preprocessed them before the assignment, i.e., tasks that required more resources than the edge server could handle were handed over to the cloud platform. When the edge servers worked together to perform tasks, the process could be divided into two major steps: the edge server to be selected was first found in scope, and then the task was logically assigned to the server to be selected.

### 4.1. Problem Definition

There are three factors to consider when finding a suitable edge server, namely the transmission distance of the task, the computing power of the edge server to be selected and its remaining computing resources.

Since the remaining computing resources of each edge server cannot be known in real time through the global network view at the initial stage of the search process, the transmission distance of the task needs to be the primary consideration at the initial stage, and the closest edge server with the lowest latency is selected, and the number of edge

servers required is determined according to the size of the task. The required number of edge servers $S_{need}$ can be obtained from the following equation:

$$S_{need} = \sup\left(R_{need}/\overline{R_{edge}}\right) \tag{1}$$

$R_{need}$ represents the compute resources required for the task and $\overline{R_{edge}}$ represents the average remaining compute resources of the edge servers within the network view. Once the corresponding number of edge servers is found, a KEEPALIVE message is sent to the specified edge server to confirm whether the edge server is still alive or not; the KEEPALIVE message is timed and has a timeout retransmission mechanism. When a reply message is received from another edge server, the current edge server stops sending KEEPALIVE messages. If a KEEPALIVE message is sent several times but no reply is received, the target edge server for the KEEPALIVE message is determined to be nonexistent or unreachable.

The reply message from the target edge server contains the existing remaining computing resources $R_{now}^i$ of the $i$th edge server. By summing the existing remaining computing resources $R_{now}^i$ of the responding edge server, the remaining computing resources $R_{temp}$ of the current edge environment can be obtained.

$$R_{temp} = \sum_{i=0}^{n} R_{now}^i \tag{2}$$

If the current resource $R_{temp}$ does not satisfy the computational resources required by the task, the search continues according to the distance and latency priority. If the current resource $R_{temp}$ satisfies the need of the task, the task allocation policy is calculated based on the obtained $R_{temp}$, and the appropriate edge server is selected for collaboration.

The selection of the optimal edge server needs to take into account the partitioning of tasks and the dependencies between subtasks. Here the dependencies between subtasks are divided into two types, namely, same-task dependencies and different-task dependencies. Same-task dependency means that there is a dependency between subtasks and they belong to the same task, while different-task dependency means that there is a dependency between subtasks but they do not belong to the same task. When subtasks are acquired in the controller, it is necessary to determine whether the current subtask will have dependencies with the next subtasks. In addition, in order to find the optimal edge server, three factors need to be considered in the task allocation.

1. Distance factor: Distance here refers to the transmission distance between the wireless sensor to be assigned the task and the edge server to be selected. Minimising the distance reduces the transmission delay of the task and the energy consumption of the sensor's emission.

2. CPU processing frequency of the edge server: The higher the operating frequency of the processing unit on the edge server, the less time a task of the same size will take to compute on the edge server and the faster it will be computed. Denoting the operating frequency of the processing unit on the edge server by $f_{edge}$, the execution time $T^{exe}$ of the task on the edge server is defined as follows:

$$T^{exe} = W/f_{edge} \tag{3}$$

3. Remaining computing resources of the edge server: The edge servers to be selected that are rich in remaining computational resources are also prioritised as the optimal edge servers to facilitate the allocation of more task-dependent subsequent subtasks. The remaining computational resources can be measured by the average system load, which can be calculated based on the average number of processes in the running queue during a given time interval. We used the following formula to define the remaining computational resources of an edge server:

$$Rm = 1 - L_{ave}/\left(N_{cpu} \times P_{cpu}^{\max}\right) \tag{4}$$

where $L_{ave}$ indicates the average number of processes in the system, $N_{cpu}$ indicates the number of CPUs, and $P_{cpu}^{max}$ indicates the maximum number of processes per CPU specified by the user, which is generally no greater than 5. The smaller the value of $Rm$, the less computing resources are left in the edge server, and when the value of $Rm$ is less than 0, the system goes into overload.

*4.2. Energy Consumption Model*

In the SDEC architecture, the underlying sensor network only undertakes data collection and data forwarding, while the computation and processing tasks are performed by the edge server. Therefore, the energy consumption of wireless sensors is divided into the sensor operation's energy consumption and the wireless transmission's energy consumption, and this paper mainly considered the wireless transmission's energy consumption of sensors. We assumed that the operational energy consumption of the sensor was stable as $E^{run}$, then according to the set of Friis transmission equations [28], the transmitting power of the wireless sensor in different environments was proportional to the distance between the transmitter and the receiver, and the ratio equation of the transmitting power $P^{send}$ to the receiving power $P^{rcv}$ was defined as follows:

$$\frac{P^{rcv}}{P^{send}} = \frac{K}{D_s^2} \tag{5}$$

where $K$ represents the influence factor in different environments and $D_s$ represents the distance between the sensor at the transmitter and the receiver. It follows that as the distance to be transmitted becomes larger, the transmit power required by the wireless sensor also needs to increase. We used $\varepsilon_{amp}$ to denote the power amplifier power consumption required to boost the transmit power of the wireless sensor, and then the transmit energy consumption formula for wireless transmission of the sensor is shown in Equation (6).

$$E^{send} = E^{elec} + \varepsilon_{amp} \times D_s^2 \tag{6}$$

$E^{elec}$ in the equation indicates the energy consumption of the circuit caused by sending and receiving data. The energy consumption equation for wireless sensors receiving data is shown in Equation (7), where the energy consumption of the sensor receiving data is independent of distance.

$$E^{rcv} = E^{elec} \tag{7}$$

When performing task offloading, the wireless sensor needs to transmit the task data to the selected optimal edge server, and the transmit energy consumption of the sensor wireless transmission is then related to the distance of transmission; since our aim was to minimise the transmit energy consumption of the sensor wireless transmission, the expression was as shown in Equation (8).

$$\min_{\varepsilon_{amp},D} E^{send} \quad s.t. D \leq D_s \tag{8}$$

*4.3. Task Allocation Issues*

The closest edge server may not have an abundance of remaining compute resources, so there is a trade-off between latency, energy consumption, the compute capacity of the edge server to be selected and the remaining compute resources. The Q-value measures the suitability of the edge server for the current edge task, with a higher Q-value indicating that the selected edge server is more suitable for the current subtask.

$$Q = \omega_1 \times f_{edge} + \omega_2 \times E^{send} + \omega_3 \times relay + \omega_4 \times Rm \tag{9}$$

The $\{\omega_1, \omega_2, \omega_3, \omega_4\}$ in Equation (9) are collectively referred to as $\omega$ values. *relay* denotes the total delay of task transmission, defined in Equation (16), and $Rm$ denotes

the remaining computational resources of the edge server, defined in Equation (4). The $\omega$ serves as the key to weighting computational capacity, delay, remaining computational resources and energy consumption, and it becomes a critical issue to find the appropriate $\omega$ both quickly and accurately.

By means of reinforcement learning, we solved the decision problem of latency, energy consumption, computational capacity of the edge server to be selected and its remaining computational resources during the edge-computing task allocation. The optimal $\omega$ choice was found by continuously measuring the edge servers to be selected based on the Q-value and updating $\omega$ based on the measurement result. Finally, the optimal $\omega$ value was substituted into the Q-value calculation formula, according to which one or more edge servers were selected for collaborative task execution. One of the measures of the optimal $\omega$ selection was the task completion time, and this paper used $relay_{fin}$ to represent the task completion time. If the task was executed on the local edge server, $relay_{fin}$ was equal to the execution time of the task in the edge cloud; while if the task was executed by multiple edge servers together, $relay_{fin}$ needed to include the time it took for the task to be passed to the edge server, the execution time of the task in the edge cloud and the time it took for the task execution result to be passed back to the local edge server. We defined the time $T^{send}$ for the task to be delivered to the edge server as:

$$T^{send} = d_i / r^{send} \tag{10}$$

where $d_i$ denotes the quantity of data to be transmitted by the $i$th sensor to the other edge servers for the edge task, and $r^{send}$ denotes the rate at which the wireless channel data are sent. Considering that after a task was processed, its data size was usually reduced compared to before processing, we calculated the sending time of the task and the receiving time of the task result separately, using $r^{rcv}$ to denote the receiving rate of the wireless channel data, and the time $T^{rcv}$ for the task result to be delivered from the other edge servers to the local wireless sensor was defined as:

$$T^{rcv} = d' / r^{rcv} \tag{11}$$

$d'$ represents the quantity of task result data received from other edge servers. The execution time of the task on the edge server is shown in Equation (3). From this, the completion time of the task $relay_{fin}$ can be obtained to satisfy the following Equation (12).

$$relay_{fin} = T^{send} + T^{exe} + T^{rcv} \tag{12}$$

If the task does not participate in coprocessing and is only executed in the local edge server, both $T^{send}$ and $T^{rcv}$ in the equation are equal to zero, at which point the task completion time is equal to the execution time of the task in the edge cloud.

In addition, in terms of task dependency, it is possible for subtasks of the same application to have task dependency requirements [4]. If subtasks with dependencies on each other are assigned to different edge servers for execution, it is also necessary to consider the time spent for the task result after the execution of the previous subtask to be delivered to the edge server executing the latter subtask, and we denoted the time delay of sending the result of the previous subtask execution by $relay_{sub}$, which was defined by Equation (13).

$$relay_{sub} = d'_{front} / r^{send} \tag{13}$$

$d'_{front}$ in the formula represents the quantity of task result data after the execution of the previous subtask. Of course, if there is no dependency between two subtasks, the size of the $relay_{sub}$ value needs not be considered. The total delay relay is defined in Equation (16) below, where $D$ is used to denote the distance between the sensor sending the current

subtask and the server receiving the subtask; $D_{sub}$ denotes the distance between the server executing the previous subtask and the server executing the current subtask.

$$relay_{pre}^{send} = relay_{sub} + D_{sub}/r^{tran} \tag{14}$$

$$relay_{next}^{send} = T^{send} + D/r^{tran} \tag{15}$$

$$relay = \begin{cases} relay_{fin} + D/r^{tran} \\ T^{exe} + T^{rcv} + \max\left\{relay_{pre}^{send}, relay_{next}^{send}\right\} \end{cases} \tag{16}$$

The definition of the total delay relay needs to take into account the transmission delay of the task, for which we used $r^{tran}$ to denote the transmission rate of the task. $relay_{pre}^{send}$ denotes the time when the result of the previous subtask is transmitted; $relay_{next}^{send}$ denotes the time when the new subtask is transmitted. When defining the total delay *relay*, if there is no dependency between the preceding and following subtasks, we do not need to consider the delivery time of the result of the previous subtask; if there is dependency between the preceding and following subtasks, we need to consider the time of the result of the previous subtask delivered to the server executing the following subtask, because before starting the execution of the task, the execution result of the previous subtask and the current subtask both must reach the edge server where the task is executed.

The algorithm first needs to define an initial $\omega_0$ for each $\omega$ and then adjust the choice of $\omega$ according to the computational load of the task, the degree of dependency and the performance of the edge server. We implemented the adjustment of $\omega$ through reinforcement learning, a process that has several key elements.

### 4.3.1. State Space $S$

The state is the information about the detection of the environment after the algorithm executes the relevant action and contains the initial state, the intermediate state and the final state. For the optimisation objectives and constraints in this paper, the state space $S$ can be defined by the variation of the task transmission delay $\gamma_{relnow}$, the variation of the task result delivery time $\gamma_{relsub}$, the variation of the task execution time $\gamma_{exe}$, and the variation of the energy consumption during the task assignment $\gamma_{energy}$, i.e., $S = \gamma_{relnow}, \gamma_{relsub}, \gamma_{exe}, \gamma_{energy}$.

The final state is the goal of the algorithm's execution, and the aim of each action's execution is to get as close as possible to the final state.

### 4.3.2. Action Space $A$

The action space $A$ contains a set of actions, which include increasing and decreasing the value of $\omega_i$, denoted by $add_{\omega_i}$ and $sub_{\omega_i}$. In addition, $nchange_i(S^-)$ in the following indicates that $\omega_i$ is adjusted in the same way as the previous action, $change_i(S^-)$ indicates that $\omega_i$ is adjusted in the opposite way to the previous action, and $k_i$ indicates that $\omega_i$ remains unchanged. The action space is defined as:

$$A = \{add_{\omega_i}, sub_{\omega_i}, nchange_i(S^-), change_i(S^-), k_i\} \\ i \in \{1, 2, 3, 4\} \tag{17}$$

The movements are initially generated randomly by the algorithm and eventually stabilise as they interact with the environment.

### 4.3.3. Reward $R$

Describing the system cost as the total duration of the task performed by the optimal edge server under the current $\omega$ selection and the energy consumption, the reward parameter $R$ is defined as follows:

$$R = \frac{(relay_{real}^{S_i^-} - relay_{real}^{S_i}) + (energy_{real}^{S_i^-} - energy_{real}^{S_i})}{relay_{real}^{S_i} + energy_{real}^{S_i}} \tag{18}$$

where $relay_{real}^{S_i^-}$ denotes the total delay of the task completion in the previous state $S_i^-$, while the total delay of the task completion in the current state $S_i$ is represented by $relay_{real}^{S_i}$. In addition, the change of energy consumption is also used as part of the reward parameter for feedback, and we used $energy_{real}^{S_i^-}$ to denote the total energy consumption under the previous state $S_i^-$ and $energy_{real}^{S_i}$ to denote the total energy consumption under the current state $S_i$.

### 4.3.4. Environment $E$

Under the SDEC architecture, the environment refers to the edge network's state information obtained by the SDN controller and the allocation of edge computing resources, such as the congestion of the network and the remaining computing resources of the edge servers. The environment information can be obtained from the SDN global view, and the intelligences are affected by the environment into different states, i.e., changes in the environment cause changes in the state. The SDN controller's role as an intelligence needs to continuously explore and interact with the environment to obtain the optimal policy for adjusting the $\omega$ value.

To find the optimal $\omega$ value, we also defined a state–action table, as shown in Table 3 below. The parameter value of action $j$ under state $S_i$ in the table is defined as $Q_{i,j}$, which is used to determine the direction of $\omega$ change. The larger the value of $Q_{i,j}$ in a given state, the more the algorithm tends to select action $j$ at state $S_i$.

The state column in Table 3 refers to the different states, while the action column contains the different actions, and the actions contain the change actions for $\omega_1 \sim \omega_4$. When the agent is in a certain state, it needs to select the action for $\omega_1 \sim \omega_4$ from the action columns, respectively. For example, at state $S_1$ in the table, the parameter value of $Q_{1,0}$ is the largest for $\omega_1$, meaning that the parameter value of the action in column 0 is the largest at state $S_1$, i.e., the parameter value of the $add_{\omega_1}$ action is the largest, at which point the algorithm tends to select the action in column 0 the most at state $S_1$, increasing the value of $\omega_1$ by $\Delta\omega$, making the change in the value of $\omega_1$ closer to the optimum. When the edge task arrives, the controller continuously adjusts $\omega$ according to the values in the state–action table and uses the adjusted $\omega$ to calculate the edge server with the largest Q-value, i.e., the most suitable edge server for the current edge task.

**Table 3.** State–action table.

| State | Action | | | | | |
|---|---|---|---|---|---|---|
| | $add_{\omega_1}$ | $sub_{\omega_1}$ | $nchange_1$ | $change_1$ | $k_i$ | $\dots$ |
| Initial state $S_0$ | 2.5 | 1 | 2 | 0.2 | 0.3 | $\dots$ |
| State $S_1$ | 2.5 | 0.4 | 2.2 | 0.1 | 1.1 | $\dots$ |
| State $S_2$ | 1.1 | 2.5 | 0.5 | 2.6 | 1.7 | $\dots$ |
| $\dots$ | $\dots$ | $\dots$ | $\dots$ | $\dots$ | $\dots$ | $\dots$ |

Moreover, in the process of continuously adjusting $\omega$, feedback is needed to update the $Q_{i,j}$ values in the $S_i$ state in the table based on the effect of the $\omega$ adjustment. We defined

the original $Q_{i,j}$ value in the table for the $S_i$ state as $Q_{i,j}^{old}$ and the $Q_{i,j}$ value in the real case as $Q_{i,j}^{real}$:

$$Q_{i,j}^{real} = R + \varphi \times \max\left(Q_{i,j}^{old}\right) \qquad (19)$$

where $\varphi$ is the decay value, and $R$ represents the reward. The current $\omega$ selection is fed back into the table through the parameter $R$ and the $Q_{i,j}$ value is updated. We defined the difference between the $Q_{i,j}$ value in the original $S_i$ state and the $Q_{i,j}$ value in the real situation as a gap, then the value $Q_{i,j}^{new}$ that needs to be updated to the state–action table can be calculated by Equation (20). It is important to note that $\alpha$ in the formula is the learning efficiency.

$$gap = Q_{i,j}^{real} - Q_{i,j}^{old} \qquad (20)$$

$$Q_{i,j}^{new} = Q_{i,j}^{old} + \alpha \times gap \qquad (21)$$

The RL-SDETA algorithm ends when the adjustment of $\omega$ enters the final state of the state–action table, when the adjustment of $\omega$ has reached its optimal value. The final state is determined when the $\omega$ value is no longer adjusted or is repeatedly adjusted around a certain value, and the determination parameters need to be set before the algorithm is executed. The significance of this algorithm is that the effect of performing an action in different states can be fed back into the state–action table by means of the reward parameter $R$. With continuous feedback, the parameter values stored in the state–action table allow the adjustment of $\omega$ values in the right direction quickly and with increasing efficiency. The process of selecting the optimal $\omega$ value for the RL-SDETA algorithm is described in Algorithm 1 and is demonstrated in Section 5 by simulation experiments where the number of lookups for the algorithm varies with the number of training rounds.

---

**Algorithm 1** Optimal value find algorithm.

---

**Input:**
 1: Give calculation amount of task
 2: Give global view information
 3: Give weight $\omega_0$ before algorithm execution
**Output:**
 4: Optimal $\omega$ value selection
 5: Initialize random data $\varepsilon$
 6: Calculate the $Q$-value of the edge server in the global view by Equation (9)
 7: **while** state $S$ != 'terminal' **do**
 8:　　Select an action $A$ from the state table corresponding to state $S$
 9:　　With probability $\varepsilon$, select a random action $A$
10:　　Otherwise, select $A = \underset{A}{\mathrm{argmax}} Q(S, A)$
11:　　Execute action $A$ and generate a new state $S_{next}$ and a reward $R$
12:　　Adjustment parameter $\omega$
13:　　**if** $S_{next}$ != 'terminal' **then**
14:　　　Calculate $C^{real}$ by $R, Q^{old}$ according to Equation (18)
15:　　**else**
16:　　　Assign the reward $R$ to $Q^{real}$
17:　　**end if**
18:　　Update the state table by parameter $gap$ according to Equation (20)
19:　　Update new state $S_{next}$ to state $S$
20: **end while**
21: **return** state table, $\omega$

---

The optimal $\omega$ value can be obtained by the above-mentioned Algorithm 1, and this $\omega$ value is substituted into the Q-value calculation formula, i.e., Equation (9). The Q-value

calculated according to the formula is used to measure the edge server, and the optimal edge server selection scheme is finally obtained. The specific process is implemented in Algorithm 2, and the edge server selection can be assigned to one or more edge servers according to the task division.

---

**Algorithm 2** RL-SDETA algorithm

---

**Input:**
  1: Give computing tasks
  2: Give global view information
**Output:**
  3: Optimal edge server selection
  4: Define initial $\omega_0$
  5: Convert global view information to array form
  6: Call Algorithm 1 with global view information in the form of index groups
  7: Calculate the $Q$-value of the edge server to be selected according to Equation (9)
  8: Sort the $Q$-values of edge servers
  9: Select the best edge server by the $Q$-value
 10: **return** Optimal server selection schemes

---

## 5. Simulation Experiment

In this section, we conduct simulation experiments on the offloading of edge task assignment under software definition. The experimental results are divided into three parts: (i) investigating the variation in the number of lookups of this reinforcement-learning-based edge task allocation algorithm; (ii) comparing the proposed RL-SDETA algorithm with random and uniform task-offloading schemes; and (iii) investigating the impact of the task's data size on task allocation performance.

### 5.1. Experimental Setup

The SDEC architecture requires an edge computing environment and we used Edge-CloudSim [29] to simulate the edge environment. The experiments simulated different numbers of edge servers running simultaneously, and the performance of the edge devices and the congestion on the links were generated randomly. The experiments used Mininet to simulate the global view transmission of the SDN's east–west architecture, and the global view information generated by the simulation was encapsulated into a configuration file whose configuration was read by the network module of EdgeCloudSim. In order to implement the software-defined reinforcement-learning-based edge task allocation algorithm proposed in this paper, the TensorFlow and Keras libraries were run in Python, running on a multicore CPU server equipped with an Intel XeonGold 5118 processor with 48 cores. In the implementation of the RL-SDETA algorithm, the global view, i.e., the task model, was used as input to the algorithm and the best server selection scheme was saved as a configuration file which was read by EdgeCloudSim to execute the scheme.

This experiment simulated different numbers of edge servers running simultaneously, and the performance of edge devices, subtask dependencies and link congestion were randomly generated. The experiments simulated global view passing of the SDN's east–west architecture with Mininet and implemented reinforcement learning algorithms using Python. Through continuous recursion, we were eventually able to obtain the optimal $\omega$ selection and apply it to the simulated environment of edge computing to verify the efficiency of task processing. The settings of the simulation parameters are shown in Table 4.

When running the RL-SDETA algorithm, as the number of training rounds continued to increase, it could be found that the number of times the optimal edge server was modified in the process of finding the optimal $\omega$ value decreased rapidly, and as the number of training rounds increased to a certain level, the number of times the optimal edge server was found stabilised at a low value. As shown in Figure 3, the number of lookups for the

optimal $\omega$ value was always maintained at a low level as the feedback algorithm continued to operate.

**Table 4.** Simulation parameters table.

| Parameter | Value |
| --- | --- |
| Number of edge servers | 10 |
| Number of tasks | 500 |
| Transmission bandwidth | 20 MHz |
| Greed | 0.9 |
| Learning efficiency | 0.8 |
| Attenuation degree | 0.9 |

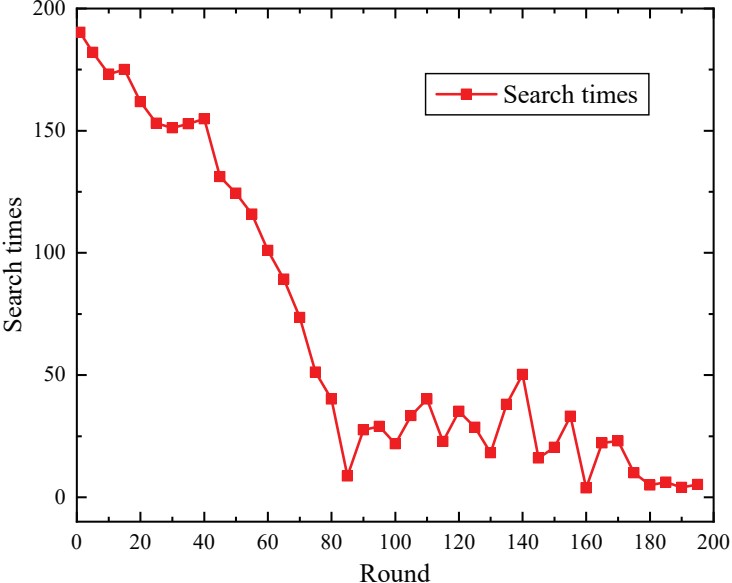

**Figure 3.** Change diagram of search times of edge server.

The reason for the rapid decrease in the number of times the value of $\omega$ needed to be modified was that as the number of training rounds increased, more and more information was fed into the state–action table. The efficiency of the controller in finding the optimal edge server was improved by continuous optimisation feedback. Once the optimal edge server was found, the controller still fed the state–action table with a positive result as a success.

It is worth noting that the number of training rounds in Figure 3 refers to the number of subtasks that arrived, with each subtask arriving to start a new round of training. When a new task arrived, the controller looked for a suitable edge server based on the size of the task and the remaining computational resources of the server; when the optimal edge server was found, the training round was completed, i.e., the number of training rounds was increased by one.

### 5.2. Performance Evaluation

In this section, we compare the RL-SDETA algorithm with the random [30] and uniform [26] task-offloading schemes. We evaluated the performance of the RL-SDETA algorithm in all aspects by comparing the computation time, latency and energy consumption of the deployed tasks.

#### 5.2.1. Contrast Programme

The random task offloading scheme refers to randomly offloading computational tasks to other edge clouds for processing or processing them locally. In our experiments, we used

a random seed that could randomly generate the number zero or one with equal probability, and then determined whether to offload the computational tasks to be processed to other edge clouds or to process them locally based on the random number.

The unified task offload scheme divided all tasks into two parts based on task size, with one part of the compute task being executed locally and the other part being executed in other edge clouds. The offloading performance was evaluated based on task duration.

5.2.2. Impact of Calculation Volume on Task Calculation Time

In the simulation experiments, we randomly generated the task size, including the amount of computation and the quantity of data for the task. For the computational volume, we used a uniformly distributed random generation; for the data volume, we used a normally distributed random generation. Varying the average computational size of the computational tasks, we compared the performance of the three algorithms when faced with tasks of different computational sizes, and the experimental results are shown in Figure 4.

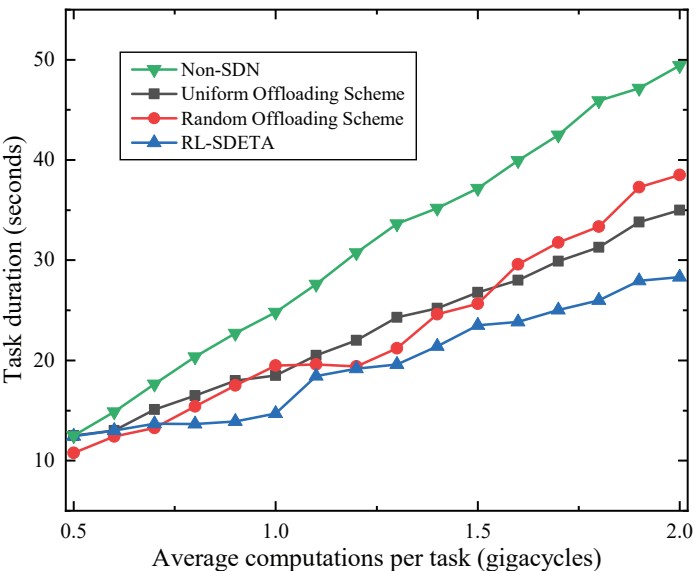

**Figure 4.** Impact of calculation amount on task duration.

As can be seen in Figure 4, the computation time of the RL-SDETA algorithm gradually became more advantageous as the average computation size of the task increased. This is the benefit of the continuous action of reinforcement learning: the influence of the actions from the environment is obtained and trained to make the correct decisions throughout the algorithm. It is worth noting that the RL-SDETA algorithm was identical to the random task offload scheme at some points because the optimal edge server calculated by the feedback algorithm happened to be the same as the randomly selected edge server, resulting in a random task offload scheme with task computation times that were consistent with or even better than the RL-SDETA algorithm, but this was not stable.

In contrast, the remaining three task-offloading schemes all had a significant advantage over schemes that did not use the SDN's global optimisation strategy. This was because with the SDN architecture, the controller could calculate the best task allocation scheme based on a global view, an advantage that offered the possibility of collaborative processing of tasks at the edge. In addition, the SDN allowed for the global optimisation of the network, which further improved the transmission rate of edge tasks.

As the RL-SDETA algorithm found the optimal edge server based on the size of the task's computation, the time advantage of the task's computation became more apparent as the average computation of the task increased. In cases where the average computation

volume of the task was small, the advantage of the RL-SDETA algorithm was not as pronounced, as even a nonoptimal edge server could complete the task quickly.

### 5.2.3. Impact of Data Volume on Task Completion Time

In this section, we measured the performance of the three algorithms when performing tasks with different quantities of data from the perspective of task data volume, and the experimental results are shown in Figure 5.

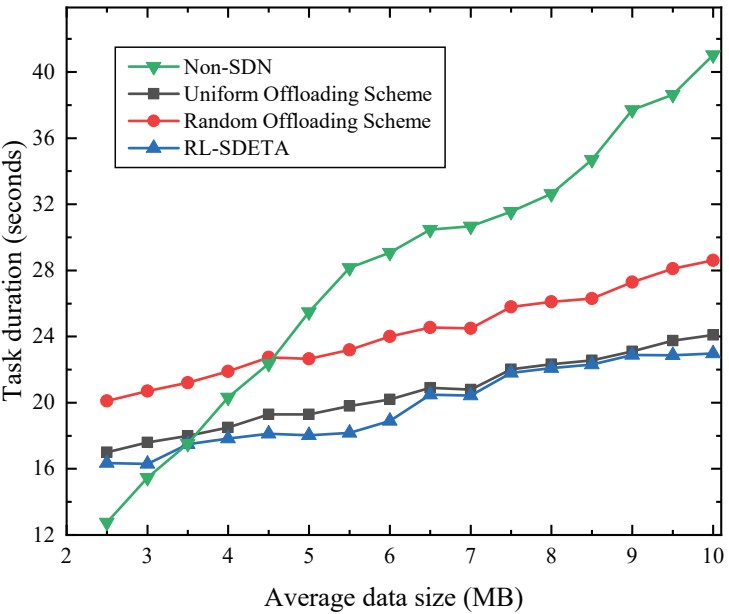

**Figure 5.** Impact of data size amount on task duration.

According to Figure 5, we can conclude that the larger the computational and data volume of the computational task, the longer the task duration. Our proposed RL-SDETA algorithm had a shorter task duration compared to the other two schemes. This was because the RL-SDETA algorithm was implemented to minimise the task duration by considering the task computation and data volume, the computational capacity of the server and the remaining computational resources. In addition, we found that the global optimisation effect of the SDN was not obvious when the task data volume was small, but as the task data volume increased, network optimisation through the SDN became necessary. This was because the SDN architecture could optimise the network congestion level when the task data volume was relatively large, and the RL-SDETA algorithm could even take the task data volume size into account in the edge server selection process.

From the above, we can conclude that the growth of the average computation and data volume of the task caused the task duration to keep growing, whereas, with the global view of the SDN, the computational resources at the edge of the network could be maximised and the medium duration of tasks were better than random and uniform task-offloading schemes.

### 5.2.4. Impact of Data Volume on Energy Consumption in Wireless Sensing Networks

By modifying the size of the average data volume, we found that the total energy consumption of the entire wireless sensing network increased as the quantity of data to be transmitted increased, which confirmed that the transmission energy consumption in a wireless sensing network is not only related to the transmission distance but also to the size of the transmitted data volume.

In this part of the experiments, we compared the difference in the total energy consumption of the wireless sensing network without using the SDEC architecture and with a path optimisation based on the SDEC architecture and using the RL-SDETA algorithm to

calculate the total energy consumption in these two different cases. From the experimental results in Figure 6, it can be seen that optimising the path in the wireless sensing network by the SDEC architecture could effectively reduce the transmission energy consumption of the wireless sensing network due to the global sensing of the SDN which could greatly reduce the path length of wireless sensors to transmit data. It can be seen from Equation (9) that the transmit energy consumption of the sensors for the wireless transmission was related to the transmission distance, so the optimisation of the path could reduce the total energy consumption in the whole wireless sensing network.

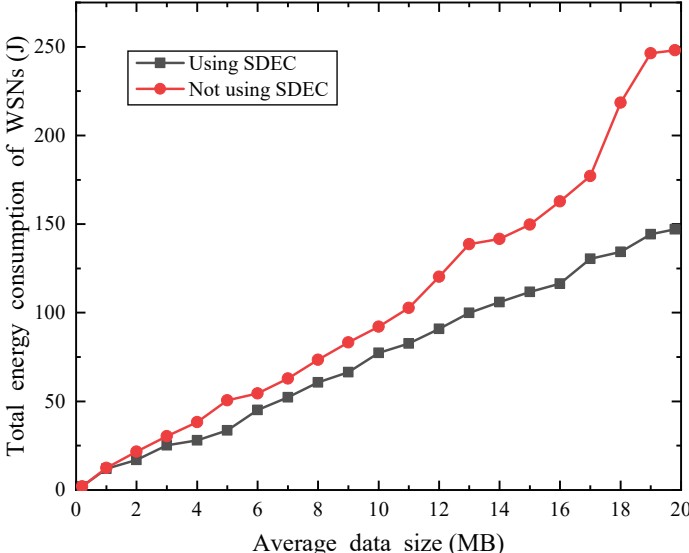

**Figure 6.** Change diagram of total energy consumption.

Using the RL-SDETA algorithm under the SDEC architecture could effectively reduce not only the energy consumption in the wireless sensing network but also the energy loss of the edge servers. Assigning tasks to different edge servers for execution could likewise solve the energy consumption problem of some energy-sensitive computing devices at the edge side.

5.2.5. Impact of The Number of Edge Servers on Task Processing

In performing this part of the performance evaluation, we allocated 500 edge tasks to different numbers of edge servers and calculated the average execution time and propagation delay of each edge task. In the edge server generation process, we randomly generated different edge server performance parameters several times to satisfy the uniform and Gaussian distributions, respectively. To consider the whole edge collaboration process comprehensively, we tested and evaluated the average propagation delay and average task execution time of multiple task-offloading schemes separately, and came to the following conclusions:

Firstly, as shown in Figure 7a, different task allocation strategies differed in their average task propagation delay under different numbers of edge servers. For example, if there was no edge collaboration, then there was no need to consider the propagation delay among edge servers; while the uniform task-offloading scheme only collaborated with the nearest edge server to process tasks and uniformly sent a portion of the edge tasks to the nearest edge server. Therefore, the propagation delay of these two schemes was the lowest. In comparison with the random task offloading scheme, the RL-SDETA algorithm had a clear advantage for the propagation delay. In the case of a large number of edge servers, the RL-SDETA algorithm considered each edge server comprehensively and selected the optimal edge server.

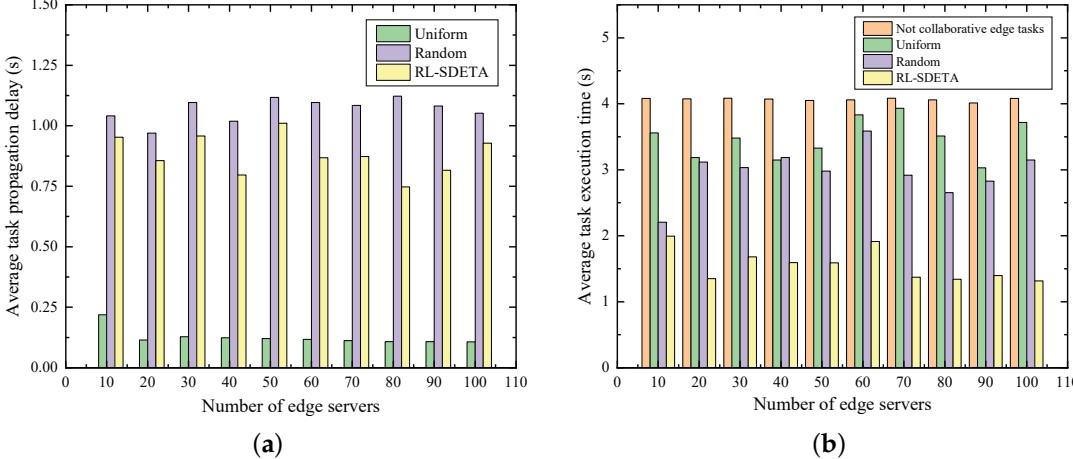

**Figure 7.** Effects of different task-unloading schemes on edge tasks. (**a**) Effect of task propagation delay. (**b**) Effect of task execution time.

Secondly, although the advantage of no edge collaboration or using only the unified task-offloading scheme was obvious in terms of the average task propagation delay, the lack of computational resources when processing tasks led to a much longer processing time for edge tasks than the other two task-offloading schemes.

Finally, as can be seen in Figure 7b, the RL-SDETA algorithm always found the most suitable solution due to learning the optimal means of measurement by means of feedback. Whether it was finding a suitable collaborative edge server or running directly at the current edge server, the RL-SDETA algorithm had a clear advantage in the task execution phase, which was the purpose of this work.

Figure 8 shows the impact of different task offload solutions on the average task processing time for each task throughout the edge task processing. The task processing time was the result of combining the sending delay, propagation delay, receiving delay and processing time of the tasks. As the number of edge servers increased, the possibility of finding a suitable edge server for a random task offloading scheme decreased significantly, and the advantage of the RL-SDETA algorithm became more obvious. The parameters in the algorithm were continuously updated with the execution of tasks, reducing the number of iterations of the search algorithm and further reducing the cost of finding the optimal edge server. From a comprehensive view, the total task processing time of the RL-SDETA algorithm was lower than that of other task-offloading schemes, which makes it more suitable for some latency-sensitive edge IoT applications.

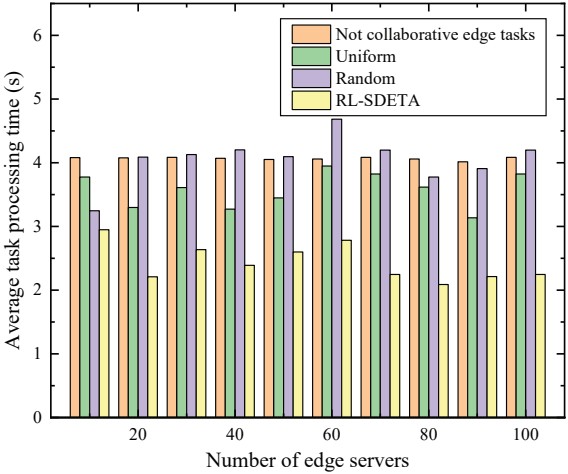

**Figure 8.** Effect of average task processing time.

## 6. Conclusions

In this paper, we proposed a software-defined edge-computing architecture, which enabled the controller to have a global view of the computing resources at the network edge. By using the global view in the controller, it was possible to assign subtasks to the optimal edge server for execution. To this end, we proposed an RL-SDETA algorithm with the goal of an optimal task assignment, and through reinforcement learning, we were finally able to obtain optimal edge server assignment schemes based on different task and data volumes. After simulation experiments, comparing the RL-SDETA algorithm with these schemes of random and uniform computational offloading, we found that the RL-SDETA algorithm worked better both for tasks with different computational volumes and for tasks with different data volumes. In our future work, we will consider the pooling of edge computing power in an SDN environment to further optimise the edge collaboration in an IoT environment.

**Author Contributions:** Methodology, T.Z. and X.Z.; software, T.Z.; validation, T.Z., X.Z. and C.W.; writing—original draft preparation, T.Z.; writing—review and editing, T.Z. and C.W.; funding acquisition, X.Z. All authors reviewed the manuscript. All authors have read and agreed to the published version of the manuscript.

**Funding:** This work was jointly supported by National Natural Science Foundation of China (grant no. 62076006), the Natural Science Research Project of Colleges and Universities in Anhui Province of China (grant no. KJ2020A0300) and the Huainan Municipal Science and Technology Project (grant no. 2021A243).

**Institutional Review Board Statement:** Not applicable.

**Informed Consent Statement:** Not applicable.

**Data Availability Statement:** Not applicable.

**Acknowledgments:** This work was supported by National Natural Science Foundation of China (grant no. 62076006), the Natural Science Research Project of Colleges and Universities in Anhui Province of China (grant no. KJ2020A0300) and the Huainan Municipal Science and Technology Project (grant no. 2021A243). The authors would like to thank the reviewers for their efforts and for providing helpful suggestions that have led to several important improvements in our work. We would also like to thank all teachers and students in our laboratory for helpful discussions.

**Conflicts of Interest:** The authors declare no conflict of interest.

## Abbreviations

The following abbreviations are used in this manuscript:

| | |
|---|---|
| IoT | Internet of things |
| MEC | Multiaccess edge computing |
| SDN | Software-defined network |
| SDEC | Software-defined edge computing |
| RL-SDETA | Reinforcement-learning-based software-defined edge task allocation algorithm |
| OXP | Open exchange protocol |
| CIDC | Communication interface for distributed control |
| WECAN | West–east control association network |
| QoS | Quality of Service |

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
