# Peer review of "Reinforcement-Learning-Based Software-Defined Edge Task Allocation Algorithm"

_electronics, doi:10.3390/electronics12030773_

Round 1

Reviewer 1 Report

Dear Authors, your topic is interesting and manuscript is well structured. However, article has high similarity i.e. 27%. Similarity with a single source is 12% that is a pre-print uploaded at researchquare but the all authors at pre-print are not included in the authors list of submitted manuscript. Why a few authors of pre-print are excluded from submitted manuscript? Further, following comments should be addressed prior to further processing of the article.

1.       Refer to affiliation: Check email address with authors’ affiliation.  

2.       Refer to abstract: Authors have proposed a software-defined edge computing architecture (SDEC) to deal with computing issue of edge servers. Why did they ignore the power od cloud computing in eth proposed architecture?

3.       Refer to references: Please confirm that referencing style is accordance with target journal.

4.       Refer to subsection 2.2: Whole subsection is composed of only one long paragraph. It dose not seem good.

5.       Refer to figure 1: Unidirectional arrows are shown to represent data flow from data plan but are missing for other data flows. Further, are these arrows unidirectional or bidirectional?

6.       Refer to whole article: Did authors consider device/controller/server heterogeneity in the the study?

7.       Refer to whole study: TCP is a connection oriented protocol. Did authors consider or plan to consider UDP as communication protocol to avoid dependence upon connection in the whole communication scenario?

8.       Refer to section 3: Insufficient detail is provided regarding establishment of TCP connection. It should be elaborated more to support easier understandability of the study back ground.

9.       Refer to subsection 4.2: Reference of Friis transmission equations is missing?

1.   Refer to algorithm 1: Caption should be “Optimal w value finding algorithm” instead of “Optimal w value find algorithm”.  

1.   Refer to subsection 5.1: Authors have used edgecloudsim, mininet and python in simulation. How did they address the compatibility of different platforms?

1.   Refer to figure 4: Graph lines of uniform and random offloading are almost parallel to each other with a small difference. How is it possible that random and uniform offloading leads to almost similar results?   

1.   Refer to figure 5: In earlier part of figure 5 the graph line of non-SDN is showing better results. How can the authors justify that performance of non-SDN is better that proposed RL-SDETA with less average data size?

1.   Refer to figure 7b: Average task propagation delay of the proposed RL-SDETA is higher than uniform offloading case. How do the authors justify it?

1.   Refer to figure 7: Recheck the caption of figure 7a and 7b. It seems swapped.

1.   For further extension and to learn about various other computing paradigms, authors are suggested to go through following study.

Waheed, A., Shah, M.A., Mohsin, S.M., Khan, A., Maple, C., Aslam, S. and Shamshirband, S., 2022. A Comprehensive Review of Computing Paradigms, Enabling Computation Offloading and Task Execution in Vehicular Networks. IEEE Access.

 Good luck

Reviewer 2 Report

The manuscript is written in a nice manner. The problem is defined well. I suggest following improvements:

1. Please add the hardware details on which simulation was run.

2. Please add some latest related references such as:

   https://doi.org/10.3390/s21051666

   https://doi.org/10.3390/electronics11152393

Reviewer 3 Report

Congratulations on this good work and very nice paper. 

Attached a reviewed version with a few typos but the content can almost be accepted in present form in my opinion. 

I would suggest a minor revision, mainly improving the litterature study because other groups have published work in the same direction, that is not mentioned.  For instance, Nikolau et al. in this paper 10.1109/TSUSC.2019.2894018

or mF2C: Towards a Coordinated Management of the IoT-fog-cloud Continuum DOI: 10.1145/3213299.3213307

  • or W.Ramírez, X.Masip-Bruin, E.Marín-Tordera, V.Barbosa, A.Jukan, G.J.Ren, O.González de Dios, "Evaluating the Benefits of Combined and Continuous Fog-to-Cloud Architectures", Computer Communications, Vol.113, pp.43-52, November 2017 

Round 2

Reviewer 1 Report

Dear Authors,

My all comments are satisfactorily addressed and I have no more comments.

Good luck